# Probing the Cellular Fate of the Protein Corona around Nanoparticles with Nanofocused X-ray Fluorescence Imaging

**DOI:** 10.3390/ijms25010528

**Published:** 2023-12-30

**Authors:** Marvin Skiba, Gabriela Guedes, Dmitry Karpov, Neus Feliu, Aitziber L. Cortajarena, Wolfgang J. Parak, Carlos Sanchez-Cano

**Affiliations:** 1Center for Hybrid Nanostructures, University of Hamburg, 22761 Hamburg, Germany; marvin.skiba@uni-hamburg.de; 2The Hamburg Centre for Ultrafast Imaging, 22761 Hamburg, Germany; 3Center for Cooperative Research in Biomaterials (CIC biomaGUNE), Basque Research and Technology Alliance (BRTA), 20014 Donostia-San Sebastian, Spain; 4European Synchrotron Radiation Facility, 38000 Grenoble, France; 5Zentrum für Angewandte Nanotechnologie CAN, Fraunhofer-Institut für Angewandte Polymerforschung IAP, 20146 Hamburg, Germany; 6Ikerbasque, Basque Foundation for Science, 48009 Bilbao, Spain; 7Donostia International Physics Center, 20018 Donostia-San Sebastian, Spain; 8Polimero eta Material Aurreratuak: Fisika, Kimika eta Teknologia, Kimika Fakultatea, Euskal Herriko Unibertsitatea UPV/EHU, 20018 Donostia-San Sebastian, Spain

**Keywords:** X-ray fluorescence imaging, nanoparticles, protein corona, intracellular fate

## Abstract

X-ray fluorescence imaging (XRF-imaging) with subcellular resolution is used to study the intracellular integrity of a protein corona that was pre-formed around gold nanoparticles (AuNP). Artificial proteins engineered to obtain Gd coordination for detection by XRF-imaging were used to form the corona. Indications about the degradation of this protein corona at a cellular and subcellular level can be observed by following the Au and Gd quantities in a time and spatial-dependent manner. The extended acquisition times necessary for capturing individual XRF-imaging cell images result in relatively small sample populations, stressing the need for faster image acquisition strategies in future XRF-imaging-based studies to deal with the inherent variability between cells. Still, results obtained reveal degradation of the protein corona during cellular trafficking, followed by differential cellular processing for AuNP and Gd-labelled proteins. Overall, this demonstrates that the dynamic degradation of the protein corona can be tracked by XRF-imaging to a certain degree.

## 1. Introduction

The use of inorganic nanoparticles (NPs) is becoming increasingly popular in the field of biomedicine, as they are promising tools capable of fulfilling multiple tasks. This includes their effective use as bright probes for fluorescence imaging [1], as well as contrast agents for X-ray imaging [2]. Additionally, NPs have been proposed as a platform for drug delivery [3], and show potential as photosensitizers for photodynamic therapy [4]. Still, it is currently difficult to predict the final benchmarking of nanoparticles as clinical agents, as their fate upon entering a biological system is not fully understood yet. This involves, for example, the endosomal escape dilemma, i.e., after their uptake by cells the NPs are located in intracellular vesicles and are not free in the cytosol. Also, their final clearance from the body may depend on the NPs’ degradation [5,6,7], and, in this way, the potential risk of long-term toxicity is a matter of concern. For a better understanding, which would help for improved NP-based drug design and realistic risk analyses, the different transformations NPs undergo in in vivo scenarios need to be quantitatively analyzed.

One of the first transformations that happen to NPs after having been introduced into an organism is the dynamic adsorption of biological entities on their surface, such as small molecules, peptides, and proteins [8]. The proteins assembled on the NP surface are also referred to as protein corona. Extensive research has shown that this protein corona may have a significant impact on the NPs’ in vitro/vivo performance, such as alteration of their biodistribution [9], cellular uptake [10], and biocompatibility [11].

While there are plenty of reports about the characterization of the protein corona in test solutions [12,13,14,15,16] after introduction into a body, analysis has been typically limited to ex vivo techniques, mainly by mass spectrometry [17]. Some non-invasive in situ and in vivo approaches have been suggested [18,19,20], but are still at the developmental stage. Optical imaging would be a handy experimental technique, as it allows us to collect subcellular information up to full-size images of organisms and even humans. Unfortunately, visible (and partly also NIR) light is scattered by tissue making it difficult to study large samples, but such a drawback has no consequences at the cellular level. With the use of fluorescent probes to label both NPs and the proteins organized around them, information about the integrity of the protein corona can be obtained. For example, light confocal microscopy showed that fluorescent labeled proteins around quantum dots (QDs) partly detached after cellular internalization [21]. A recent experiment using correlative light confocal and electron microscopy suggested the intracellular degradation of a corona of fluorescent labeled plasma proteins formed around polystyrene nanoparticles. This process occurred through endosomal pathways in a timeframe of hours, involving fast exocytosis of nanoparticles while labeled proteins stayed inside cells longer [22]. Remarkably, the use of X-ray fluorescence instead of optical fluorescence might be potentially helpful to image the protein corona also in samples at the organ/organism level, as scattering of X-ray emissions in tissue can be subtracted by appropriate background-reduction algorithms [23]. We thus investigate in this work the potential of X-ray fluorescence imaging (XRF-imaging) for analysis of the protein corona, at the cellular level, which potentially also could be extended in the future to bigger-sized in vivo samples.

XRF-imaging uses X-rays to excite the sample and collect distinct elemental-specific X-ray fluorescence spectra that can be recorded pixel by pixel to obtain elemental maps [24]. This provides direct detection of NPs and other metal-based systems within biological samples. Moreover, recent developments permit us to focus synchrotron radiation to a full width at half maximum (FWHM) of 20 nm [25], thus allowing us to acquire XRF-imaging with subcellular resolution and high elemental sensitivity. Still, XRF-imaging also has some drawbacks. For example, the acquisition of X-ray fluorescence maps from light elements, such as carbon, oxygen or nitrogen (main components of proteins), is hindered, as they suffer from high self-absorption in cellular or tissue samples [26]. Furthermore, XRF-imaging cannot discriminate between emissions originating from intracellular structures or biomolecules and those from an exogenous protein corona, unless those proteins contain heavy elements with low bio-occurrence in their structure.

Herein, we present a proof-of-concept experiment validating the use of XRF-imaging to study the intracellular fate of the protein corona formed around AuNP with engineered consensus tetratricopeptide repeat proteins (CTPR) [27], displaying specific metal-binding sites [28] for Gd labeling as a model protein, that were preadsorbed to the NPs before their internalization by cells. Therefore, it is possible to follow the integrity of the corona by assessing the colocalization between Au (NPs) and Gd (as a tracer for the protein corona) in 3T3 murine cells cryo-preserved at different times after the internalization of the AuNP@CTPR-Gd system. Overall, this work demonstrates that, in principle, biologically relevant information on the fate of protein corona in cells could be obtained if large cell populations are probed using the same experimental approach.

## 2. Results and Discussion

### 2.1. Gold Nanoparticles Used for Treatment

Gold nanoparticles were synthesized following a previously described method [29]. Their surface was then modified with thiolated polyethylene glycols bearing terminal amine groups to enhance their possibilities of forming a protein corona with negatively charged Gd labeled CTPR (zeta potential of −24.9 ± 9.7 mV). The resulting AuNPs were characterized by UV-Vis absorbance spectroscopy, zeta potential analysis, and transmission electron microscopy (TEM). Success of the surface modification was detected by a slight shift in the wavelength of the surface plasmon resonance of the gold nanoparticles (caused by a change of the surrounding refractive index) observed in their UV-Vis absorption spectra (Appendix A), and supported by zeta potential changes (from −36.9 ± 1.3 mV for citrate AuNP to +27.3 ± 2.7 mV for PEG AuNP). Additionally, negative staining (with Uranyl acetate) TEM showed a narrow size distribution for the AuNP (Appendix A), with an average core diameter of d_c_ = 11.9 ± 0.7 nm (*N* = 121). The diameter of the whole NPs, including the organic PEG shell, was extended d_cs_ = 32.5 ± 4.9 nm (*N* = 189). The formation of the protein corona of CTPR-Gd about the final particles was confirmed by gel electrophoresis (Appendix A). Moreover, an indirect method combining low-speed centrifugation, Bradford assay, and ICP-MS, for protein and metal concentration quantification, respectively, revealed the presence of a hard corona containing around 130 proteins per AuNP.

Finally, viability assays on mouse embryonic fibroblasts (3T3) cells revealed that pegylated AuNP, AuNP@CTPR-Gd, and free Gd labeled CTPR caused no acute toxicity up to 10 µg/mL Au or 2 µM protein (Appendix A). Notably, the choice of 10 µg/mL Au treatment for following XRF-imaging experiments and ICP-MS uptake experiments was intentionally made to ensure suitability for detection by XRF-imaging without overloading the cells with particles and thus create random overlap of the different components. Remarkably, a slight increase on cell viability as observed at higher CTPR-Gd concentrations might be due to possible interferences on the used 3-(4,5-dimethylthiazol-2-yl)-5-(3-carboxymethoxyphenyl)-2-(4-sulfophenyl)-2H-tetrazolium (MTT) assay caused by the protein, as reported for different polypeptides before [30]. Further, cellular uptake experiments (Appendix A) determined that each individual cell internalized an average of 0.17 ± 0.04 pg Au and 0.22 ± 0.06 pg Gd after 24 h of treatment with AuNP@CTPR-Gd (*C*_Au_ = 10 µg/mL).

### 2.2. X-ray Fluorescence Imaging

To study the dynamic behavior of the protein corona after cellular internalization, we exposed 3T3 cells to AuNP@CTPR-Gd for 24 h (*C*_Au_~10 µg/mL), followed by different incubation times t_inc_ in particle-free medium (0, 30, 60 or 120 min). Cells were cryopreserved and analyzed with XRF-imaging under close-to-native conditions at the ID16A cryo-nanoprobe beamline at the ESRF. As a control, untreated cells were prepared and analyzed in the same way. Initially, coarse scans were conducted (400 nm × 400 nm step size and 100 ms dwell time) to select 3T3 cells adequate for imaging. Information about the cell state was obtained based on K K_α_ emission maps, which indicated which cells were alive before fixation by plunge freezing and maintained their integrity during the process. Moreover, both K and Zn K_α_ emissions were used throughout the experiments to identify the cellular and nuclear outlines (respectively). Subsequently, at least three cells were imaged by XRF-imaging for each group (controls, or AuNP@CTPR-Gd exposure for 24 h followed by 0, 30, 60, and 120 min of NP-free incubation), with a lateral resolution of 70 nm × 70 nm. In total, 16 cells were imaged for this study. As expected, no Au- or Gd-L X-ray emission peaks were observed in the X-ray fluorescence (XRF) spectra obtained from untreated 3T3 controls, but were clearly visible in those from cells exposed to AuNP@CTPR-Gd (Appendix A). This allowed for obtaining maps of Au and Gd as unique tracers for the NP cores and the protein corona (respectively) for each sample; see, e.g., Figure 1. Pseudo-colored images of K, Zn, Au, and Gd for all measured high-resolution cells are listed in the SI section (see Appendix A).

An initial evaluation shows the overlap between the L_α_ emissions from Au and Gd occurring inside cells. This finding is in accordance with the expected simultaneous uptake mechanism for the hybrid construct AuNP@CTPR-Gd, i.e., the preformed protein corona is endocytosed together with the NP cores. The granular structure of the NP distribution is in agreement with endocytotic uptake [31,32,33,34]. Both Au and Gd elements appear to be mostly accumulated in multiple small areas (which is clearer in the Gd maps), which are attributed to lysosomes/endosomes [35].

### 2.3. Analysis in Whole Cells

The absolute amount of Au and Gd and their ratio was determined for the whole cellular area, but a clear tendency showing the decrease or increase as a function of the incubation time could not be observed (Figure 2). Nevertheless, a great degree of cellular variability in the Au/Gd ratio could be observed in the small population of cells studied, making it difficult to assess the results obtained. Yet, although the initial molar ratio of Au per Gd in AuNP@CTPR-Gd hybrids before they were added to cells was *c*_Au_/*c*_Gd_ = 26.8, the values obtained from our XRF-imaging measurements after NP internalization by cells were around *c*_Au_/*c*_Gd_ = 2. Such a low ratio was supported by ICP-MS-based uptake experiments, which displayed similar quantities of Au and Gd per cell and a ratio of *c*_Au_/*c*_Gd_ = 0.9. In general, this might indicate that the AuNP taken up during the 24 h treatment with AuNP@CTPR-Gd are excreted faster than the Gd labeled proteins, which is in line with fluorescence-based degradation studies and also experiments using polystyrene NPs [21,22].

For further colocalization analysis, the elemental concentrations (ng/mm^2^) of the XRF-imaging maps acquired were first converted into nmol/mm^2^ by dividing the value of each pixel by the molecular weight of the element of interest. This allows for a more meaningful data interpretation by comparing the relative amounts of Au and Gd atoms. Masks marking the outline of each individual cell were generated from the potassium XRF-imaging maps, and the colocalization between Au and Gd was calculated inside those areas to avoid any possible interference from the background signal. The Pearson’s correlation coefficient r and the Manders’ coefficients *M*_1_ and *M*_2_ were determined for whole cells (Figure 2). For all colocalization analysis, channel 1 and thereby, *M*_1_ is based on Au, whereas channel 2 (*M*_2_) defines the contribution from Gd.

Overall, all correlation coefficients show values > 0.6, indicating that Au and Gd are highly colocalized inside cells. Still, Pearson’s correlation coefficient r shows a slight decrease in its mean value within the first 60 min. This might imply a possible decomposition of the protein corona whereby proteins (as indicated by Gd) are removed from the surface of the NPs, as in that case the colocalization between both elements should decrease. Still, these variations are not statistically significant, proven by Student’s *t*-test (*p* > 0.05). This might be caused by the overall small populations in these studies (i.e., three cells per condition) and the huge intrinsic cellular variability observed, as more clearly seen by displaying the individual values in the form of a box plot (Appendix A), which does not allow us to establish proper statistics.

In addition, the mean values of *M*_1_ as well as *M*_2_ are nearly constant, characterized by a large error, and thus tolerate no accurate discussion of the results.

### 2.4. Analysis Inside Cellular Vesicles

The same colocalization analysis was performed at the small circular areas inside cells, where most Au and Gd were found. These intracellular vesicles are most likely endosomes and lysosomes linked to the internalization process of the NPs. Previous experiments suggested that protein corona around NPs might be degraded in endosomes, during their maturation into lysosomes [21,22,36]. For the analysis, at least 20 of those areas were automatically selected based on the Gd signal for each image using Fiji software V. 1.53c. This was carried out in order to avoid human-related bias in the selection of the regions of interest (ROI). All of the regions selected had a circularity ≥ 0.7 and showed a diameter smaller than 1 µm (area < 0.79 µm^2^), as expected from cellular endosomes or lysosomes. Representative images showing the ROIs selected from cells exposed to AuNP@CTPR-Gd for 24 h and different subsequent incubation times t_inc_ in NP-free medium are shown in Figure 3. Images showing the ROIs selected from all cells analyzed can be found in Appendix A.

The Au to Gd ratios *c*_Au_/*c*_Gd_ found in vesicles were always smaller than those obtained when whole cells were considered (Figure 2 and Figure 4). Therefore, a substantial fraction of intracellular Gd must remain located inside endosomes/lysosomes. Moreover, our analysis also showed that the molar ratio *c*_Au_/*c*_Gd_ inside vesicles decreased over time, from an initial value of 1.51 ± 0.02 in cells (t_inc_ = 0), to 1.32 ± 0.01 after t_inc_ = 120 min (*p* = 0.014). By calculating the elemental amount of Au and Gd in all the areas stated as vesicles, also a slight decrease in Au can be seen over time, whereas the amount of Gd stays at almost the same level (Figure 4). These results are significant and seem to support the degradation of the protein corona inside the vesicles, followed by a faster removal of AuNP from them. Remarkable, small vesicle-like areas containing only Au are found close to the nuclei in all cells studied (see Figure 1). Thus, vesicles seem to maintain the same quantity of Gd labeled proteins, but reduce their amounts of AuNP over time leading to decreased Au/Gd ratios. This corresponds to previous studies, where it was shown that a core size d_c_ ≈ 12 nm still can be endocytosed sufficiently [37,38,39,40].

Unfortunately, we cannot assess the degree of degradation of the protein corona in vesicles between the cell uptake of intact AuNPs@CTPR-Gd hybrids and the separation of AuNPS and CTPR-Gd at nuclear areas. Still, differential cellular processing for AuNPs and CTPR-Gd is only observed in areas close to the nuclei, which might indicate that cells cannot treat particles and proteins as individual entities. Thus, some of the initial protein corona might still exist during the intermediate stages of cellular trafficking.

An analysis of the Au-Gd colocalization in vesicle areas is shown in Figure 5. A small decrease in the values of PCC (r) and Manders’ coefficient 1 (*M*_1_; the relative proportion of Au overlapping with Gd) could also be detected over time, supporting again the possible decomposition of the protein corona inside vesicles, e.g., endosomes or lysosomes, suggested by the changes in the Au/Gd ratios. However, no change was observed for Manders’ coefficient 2 (*M*_2_; relative proportion of Gd overlapping with Au) at different incubation times. Previous reports proposed that after degradation, NPs are separated from the detached proteins that had originally formed the corona and are trafficked into vesicles that excrete them from the cell, or other organelles where they might accumulate for some time [41]. This could explain the decrease in r and *M*_1_ during the incubation period, as AuNP would leave the initial endosomes/lysosomes, and no new AuNP@CTPR-Gd would be internalized by cells to maintain the correlation between Au and Gd in vesicles. Instead, CTPR-Gd proteins would stay inside the initial vesicles, as proteins desorbed from the NPs’ surface during this time scale might not be able to escape, so (keeping the spatial resolution of 70 nm × 70 nm in mind) no observable change might be detected in *M*_2_.

Nevertheless, again these results were obtained from small cell populations (i.e., three cells per condition), and thus differences in colocalization coefficients could not be considered significant (Student’s *t*-test: *p* > 0.05) due to cellular variability, making it impossible to obtain relevant conclusions.

### 2.5. Analysis Dependent on Distance to Nucleus

During cellular internalization, AuNP@CTPR-Gd would be trafficked by maturing endosomes from distal regions of the cell into areas closer to the nucleus. As such, degradation of the protein corona during cellular transport might be observed as possible changes in the relative accumulation of the AuNP and CTPR-Gd proteins in regions distal and closer to the nucleus. Therefore, each cell image was divided into a number of ROIs, defined as 1 µm width bands with increasing separation from the cellular nucleus. Since cells show a variety of sizes, no ROIs were defined farther than 6 µm from the nucleus, as this allowed the study of at least three cells for every sample group. Representative images showing the regions of interest selected from cells exposed to AuNP@CTPR-Gd for 24 h and different following incubation times in NP-free medium are shown in Figure 6. Images showing the ROIs selected from all analyzed cells can be found in Appendix A. As images are two dimensional, it should be noted that there is no way to determine whether an element is found inside or outside the nucleus.

The absolute amount (Appendix A) and molar ratios of Au to Gd (Figure 7) were determined for each of these areas with increasing distance d_nuc_ to the cellular nucleus. Remarkably, the Au/Gd ratios increased from areas in the outside of the cell towards those in the proximity of the nucleus. Such spatial dependence seems to support the hypothesis that the initial protein corona is removed from the NP surface during cellular trafficking, leading to different cellular processing of AuNP and CTPR-Gd proteins. Moreover, as the Au to Gd ratio is normally smaller in areas defined as vesicles than in the proximity of the nucleus, our analysis also suggests that some AuNP must be capable of escaping endocytic vesicles and/or be transported or accumulated into different organelles. This can explain the presence of vesicle-like areas containing only Au close to the nuclei in all cells studied (see Figure 1).

It has been reported that vesicles containing NPs are able to fuse with organelles related to the cellular secretion pathways, e.g., Golgi apparatus or rough endoplasmic reticulum, whereas the following fate is not fully revealed [41]. These organelles are close to the nucleus. Interestingly it was also shown that 12 nm AuNPs modulate the function of the endoplasmic reticulum [42]. Moreover, supporting our findings, electron microscopy studies examinating the localization of 12 nm AuNPs show an accumulation of most NPs inside vesicles in close proximity to the cellular nucleus [43].

However, although the highest Au/Gd ratios within cells are always found near the nuclei for all time points studied (Figure 7), there is a significant (Student’s *t*-test: *p* < 0.05) reduction over time on the relative amount of Au to Gd in those areas. The Au/Gd ratios near the nuclei go down from *c*_Au_/*c*_Gd_ = 3.3 ± 0.5 at 24 h exposure without additional incubation time in NP-free medium, to 1.9 ± 0.3, 2.0 ± 0.3 or 2.8 ± 0.4, for 30, 60, or 120 min incubation time, respectively. As such, it might be possible that over 24 h of cellular uptake of AuNP@CTPR-Gd cells reached a steady state with a slightly higher Au amount near the nucleus and a fully homogenous Gd distribution in the cell (Figure 8). Yet, 30 min after stopping the uptake of new NPs, the quantities of both Au and Gd increased in the nuclear area (Figure 8), while Au/Gd colocalization decreased as shown by their r, *M*_1_, and *M*_2_ coefficients (Appendix A). This indicates separate intracellular processing for AuNP and CTPR-Gd proteins. Furthermore, during the following t_inc_ = 90 min of incubation, the amount of Au and Gd is redistributed all over the cell, possibly preparing AuNP for efflux and CTPR-Gd proteins for further digestion [22].

Unfortunately, clear evidence of this process could not be drawn due to the limited sample population causing high standard deviations, and thus our arguments remain speculative.

### 2.6. Post Hoc Sample Size Estimation

The large cell variability and small sample populations have been an issue throughout this study, hindering the capacity to obtain meaningful conclusions from some of the analyses performed on the data. Unfortunately, this problem is not unique to this experiment. Currently, most XRF-imaging-based experiments intending to obtain subcellular information need to settle for long acquisition times per sample to achieve the spatial resolution required. Thus, reducing the maximum size of the sample population is possible to achieve, and the scope of the study itself.

As a theoretical exercise, we estimated the sample size required to achieve meaningful results on the intracellular degradation of the protein corona in a hypothetical future experiment. To achieve this, we used a statistical approach, commonly used to compute group sizes for clinical trials, that was performed using the G*Power software V. 3.1.9.7. [44]. As an example, the PCCs (r) obtained from our current experiment were projected as a normal distribution with a FWHM of their standard deviations. From this, a post hoc analysis calculated the individual power (Cohen’s *d*) as a measure for the difference in two populations [45]. After extracting all *d* values, an a priori simulation can calculate the minimum sample size for each incubation time to have a significant difference. In the case of the whole cell area analysis, these calculations led to the finding that at least 6 cells needed to be measured in the most different case, 0 versus 60 min incubation time, 13 cells by comparing 0 versus 30 min incubation time, and, in the most similar scenario, 0 versus 120 min incubation time, 20 cells are required for each study group.

Unfortunately, currently, it might be challenging to obtain images of such a large number of cells within the timeframe of a normal experiment using the setups and capabilities of nanoprobe beamlines open at this time [46,47,48,49].

## 3. Materials and Methods

### 3.1. Nanoparticle Synthesis

Au NPs with a core diameter of 12 nm were synthesized following a previously published experimental protocol [29]. A detailed description can be found in the Appendix A.

### 3.2. Surface Modification of Nanoparticles

AuNPs were functionalized with thiolated polyethylene glycols (PEG) bearing terminal amine groups, which generated NPs with positively charged surfaces. A detailed experimental description of surface modification can be found in the Appendix A, alongside with synthesis and characterization of the CTPR-Gd protein used (Appendix A).

To mimic a biological protein corona, AuNPs used to treat cells were incubated with 900 equivalents of CTPR-Gd protein for 2 h at 37 °C, following 3 washing steps by centrifugation (25 krcf, 30 min) replacing the supernatant with Alpha-modified eagles’ medium (AMEM), without fetal bovine serum (FBS), to remove loosely bound proteins. Indirect measurement by inductively coupled plasma-mass spectrometry (ICP-MS/Thermo Fisher iCap-Q, Waltham, MA, USA) of Au and Gd revealed that 210 proteins were attached to each AuNP after these three washing steps.

Characterization of the NPs (TEM/JEOL JEM-2100F UHR, Akishima, Japan, UV-vis spectroscopy/Agilent Varian Cary 5000, Santa Clara, CA, USA), along with viability studies, can be found in the SI.

### 3.3. X-ray Fluorescence Imaging Sample Preparation

Mouse embryonic fibroblasts (3T3-cells) were cultured onto silicon nitride membranes for nanofocused synchrotron XRF-imaging experiments following methods adapted from previously published protocols [50]. The cells were detached from the culture flasks and diluted with AMEM culture media to 50,000 cells/mL. Then, one drop (10 µL) each was placed onto pre-treated silicon nitride membranes, which were fixed to the bottom of 6-well plates using UV-sterilized double-sided adhesive tape. The use of double-sided adhesive tape prevents the membrane from floating around in the well, thereby minimizing the risk of breakage during the experimental procedure. After two hours of incubation (37 °C, 5% CO_2_), 2 mL AMEM media was carefully added, and the plate was placed in an incubator overnight. On the following day, the medium was removed and replaced with fresh AMEM medium, without FBS, for the control group, or AMEM, without FBS, containing AuNP@CTPR-Gd (with an elemental concentration of Au as determined by ICP-MS of *C*_Au_~10 µg/mL), respectively. After 24 h of treatment with AuNP@CTPR-Gd, cells were washed with 2 mL phosphate-buffered saline (PBS) and kept in 2 mL fresh growth medium (no AuNP) for different incubation times (in this work t_inc_ = 0, 30, 60, and 120 min). Afterwards, the membranes were taken out, immersed in 150 mM ammonium acetate buffer (pH 7.1), blotted with filter paper, and immediately plunge frozen in liquid ethane. Samples were then transferred into in-house designed and 3D-printed sample holders and kept under cryogenic conditions until measurements were performed.

### 3.4. Synchrotron X-ray Fluorescence Imaging

All synchrotron experiments were performed at the beamline ID16A at the European Synchrotron Radiation Facility (ESRF, Grenoble, France). The XRF-imaging images of individual cells were acquired under cryogenic conditions using two six elements silicon drift diode detectors with an area of 3 cm^2^ at a distance of 3 cm from the sample. The incident beam energy was fixed to 17 keV (with a photon flux of 1.55·10^11^ ph/s) and focused to 48.6 nm × 41.6 nm (FWHM). Fine cellular mapping was conducted using a 70 nm × 70 nm step size with a 50 ms dwell time. Combined emission spectra were fitted and analyzed using the PyMCA software V 5.6.5. [51]. Elemental concentrations were examined assuming a 6 µm thick matrix with a density of 1 g/cm^3^.

### 3.5. Colocalization Analysis

The obtained elemental maps were investigated using a pixel-by-pixel-based analysis. In detail, Pearsons’ correlation coefficients (r, *PCC*) and Manders’ coefficients (*M*_1_*, M*_2_) were calculated by the ImageJ-Coloc2 plugin [52]. A theoretical description of the methodology can be found in the Appendix A. In short, the elements representing the nanoparticle (Au) and protein corona (Gd) were studied regarding their co-occurrence, despite individual intensity (*r*) or minding their intensities (*M*_1_, *M*_2_) in single pixels. Every colocalization analysis was verified by Costes’ significant test (*p* ≥ 0.95).

## 4. Conclusions

In this study, pegylated AuNP with a core diameter of 12 nm and a preformed corona of engineered proteins labeled with Gd surrounding them were probed inside mammalian cells using synchrotron nanofocused XRF-imaging. Our analysis shows that the colocalization between Au and Gd on whole cells decreased over time, although to a very limited degree, while a meaningful reduction in the Au/Gd ratio inside vesicles was also observed over time. Additionally, higher Au/Gd molar ratios were found in areas close to the nuclei than in those far from it, but those differences across cells were reduced after they stopped internalizing AuNP@CTPR-Gd. Data interpretation was complicated due to large cell-to-cell variance. Still, our results seem to indicate degradation of the protein corona during the trafficking (and endosomal maturation) of AuNP@CTPR-Gd hybrids after vesicle-related cellular uptake at the outer membranes. Moreover, our data also suggests different cellular processing for AuNP and CTPR-Gd proteins, as supported by the absolute decrease in the amount of Au inside the vesicles over time.

Overall, this work demonstrates that XRF-imaging can be used to study the intracellular fate of the protein corona, and might be a valid tool to investigate other dynamic processes inside cells. Nevertheless, it is worth mentioning that the current technical capabilities of most hard X-ray nanoprobes impose long acquisition times to obtain single images with subcellular resolution, limiting the maximum sample size of the cell populations studied. It is possible to obtain some meaningful results with such a sample size (as we did), but in general, it makes it difficult to overcome the inherent cell-to-cell variance and can hamper the application of XRF-imaging to other cellular processes. Therefore, new technical advances allowing faster acquisition times of images with subcellular resolution would be beneficial to extend the use of XRF-imaging for the study of dynamic biological processes at the cellular level.

## Figures and Tables

**Figure 1 ijms-25-00528-f001:**
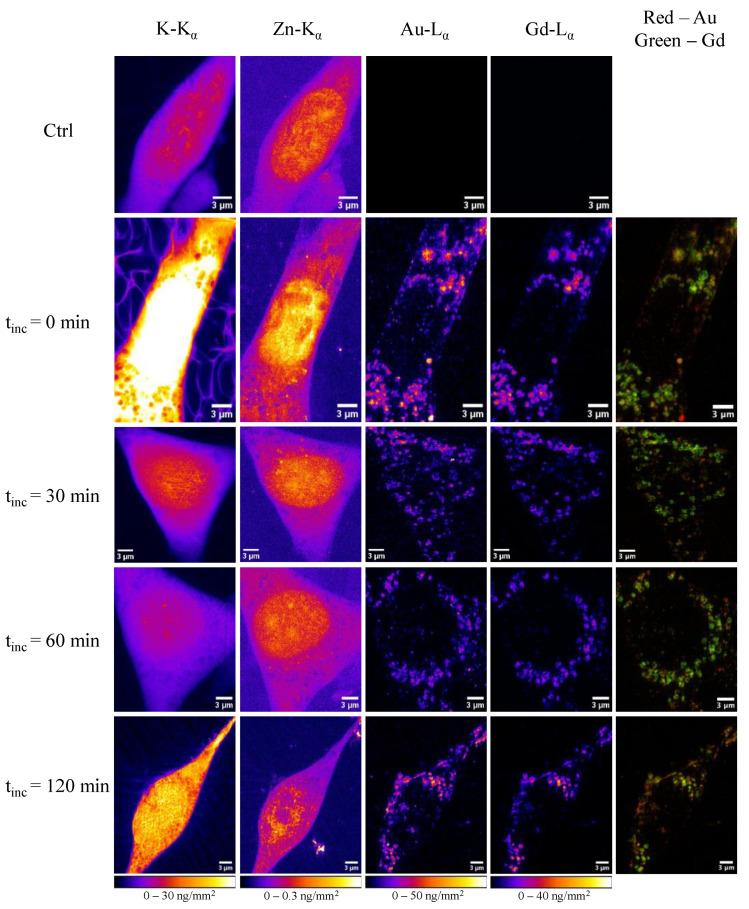
Pseudo-colored images of 3T3 cells acquired by XRF-imaging. The signals originated from fitted: K K_α_ emission (first column), Zn K_α_ emission (second column), Au L_α_ emission (third column), and Gd L_α_ emission (fourth column). Fifth column shows a merged image indicating a high colocalization, whereas red originates from Au emission and green from Gd emission. The cells depicted in the first row are control cells without NP treatment. The following lines show one representative cell after 24 h of exposure with AuNP@CTPR-Gd (*C*_Au_~10 µg/mL) and different incubation times t_inc_ in NP-free medium as indicated on the left. The scale bars indicate 3 µm.

**Figure 2 ijms-25-00528-f002:**
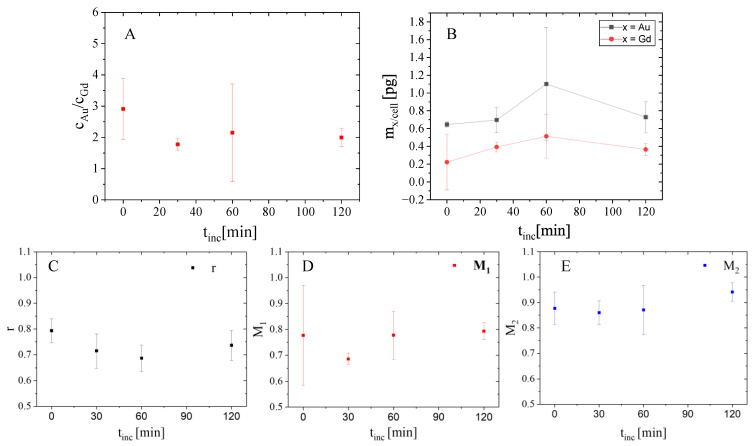
Molar ratio *c*_Au_/*c*_Gd_ (**A**) and total elemental amount (**B**) of Au and Gd (m_Au/cell_, m_Gd/cell_) in whole 3T3 cells exposed to AuNP@CTPR-Gd (*C*_Au_~10 µg/mL) for 24 h, followed by different incubation times (t_inc_ = 0, 30, 60, and 120 min) in NP-free medium. Lower row illustrates consecutive colocalization between Au and Gd in these cells. The graphs show the Pearson’s correlation coefficient r (**C**), the Manders’ coefficient *M*_1_ ((**D**) relative proportion of Au overlapping with Gd), and the Manders’ coefficient *M*_2_ ((**E**) relative proportion of Gd overlapping with Au).

**Figure 3 ijms-25-00528-f003:**
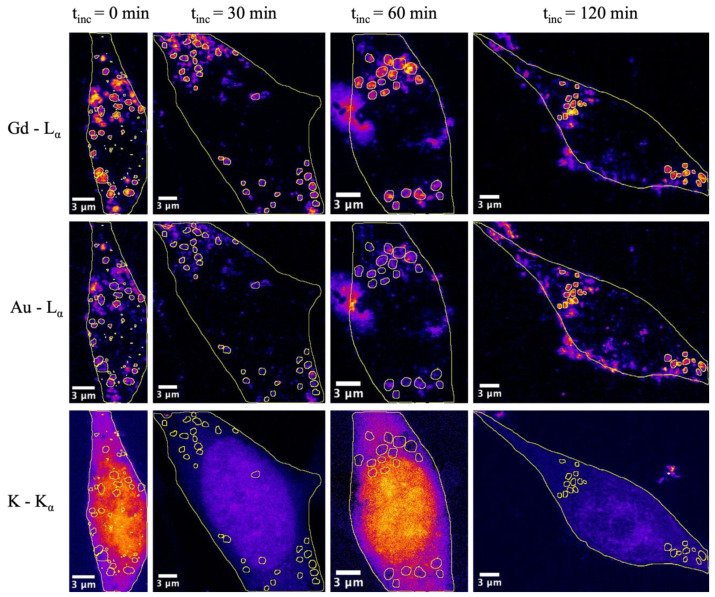
Pseudo-colored images acquired by XRF-imaging from 3T3 cells exposed to AuNP@CTPR-Gd for 24 h (*C*_Au_~10 µg/mL) after subsequent incubation times t_inc_ in NP-free medium: t_inc_ = 0, t_inc_ = 30 min, t_inc_ = 60 min, and t_inc_ = 120 min. The signals originated from the Gd-L_α_ emission (first row), Au-L_α_ emission (second row) or K-K_α_ emission (third row). Regions of interest (ROI) in yellow stated as intracellular vesicles are defined for nearly spherical areas, showing a circularity > 0.7 and an area smaller than 0.79 µm^2^. The scale bars indicate 3 µm.

**Figure 4 ijms-25-00528-f004:**
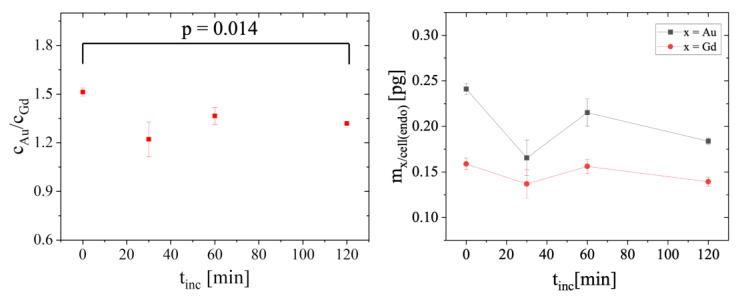
Molar ratio *c*_Au_/*c*_Gd_ of Au/Gd (**left**) and the total amount (pg/cell) of Au and Gd (**right**) inside structures stated as vesicles (at least 20 per cell) m_X/cell(endo)_ (X = Au, Gd) in 3T3 cells exposed to AuNP@CTPR-Gd (*C*_Au_~10 µg/mL) followed by different incubation times in NP-free medium (t_inc_ = 0, 30, 60, and 120 min).

**Figure 5 ijms-25-00528-f005:**
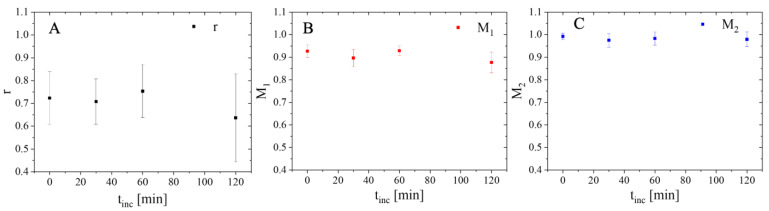
Colocalization between Au and Gd inside structures stated as vesicles (at least 20 per cell) in 3T3 cells exposed for 24 h to AuNP@CTPR-Gd (*C*_Au_~10 µg/mL), followed by different incubation times in NP-free medium (t_inc_ = 0, 30, 60, and 120 min). Graphs show Pearson’s correlation coefficient r (**A**), Manders’ coefficient *M*_1_ ((**B**) relative proportion of Au overlapping with Gd), and Manders’ coefficient *M*_2_ ((**C**) relative proportion of Gd overlapping with Au).

**Figure 6 ijms-25-00528-f006:**
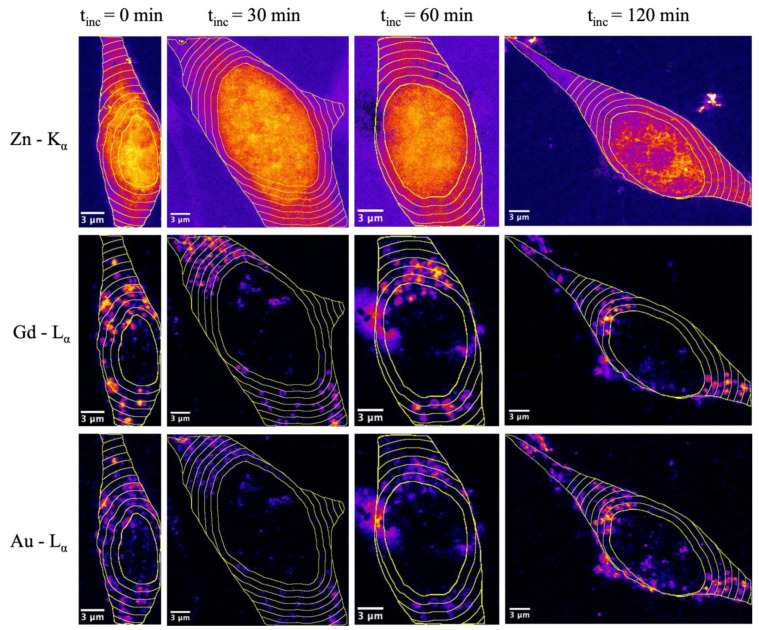
Pseudo-colored images acquired by XRF-imaging from 3T3 cells exposed to AuNP@CTPR-Gd for 24 h (*C*_Au_~10 µg/mL), followed by incubation in NP-free medium for the time t_inc_: t_inc_ = 0, t_inc_ = 30 min, t_inc_ = 60 min, and t_inc_ = 120 min. The signals originated from the Zn K_α_ emission (first row), Au L_α_ emission (second row), or Gd L_α_ emission (third row). ROIs are generated by defining the cellular nucleus as the center (identified from the Zn K_α_ emissions maps), the outer cellular membrane as border (identified from the K K_α_ emissions maps), and sectioned into rings with a width of 1 µm. The scale bars indicate 3 µm.

**Figure 7 ijms-25-00528-f007:**
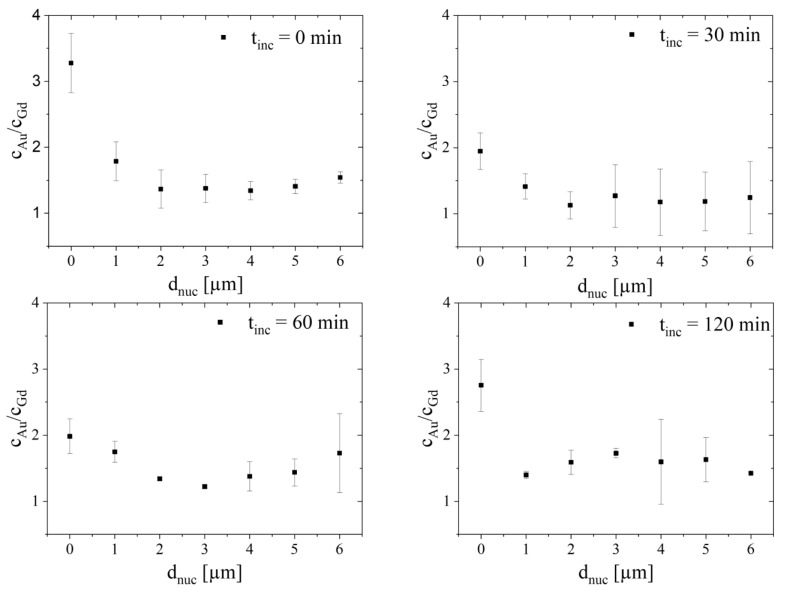
Molar ratio *c*_Au_/*c*_Gd_ of Au/Gd according to the distance d_nuc_ to the cellular nucleus in 3T3 cells exposed to AuNP@CTPR-Gd (*C*_Au_~10 µg/mL), followed by different incubation times in NP-free medium (t_inc_ = 0, 30, 60, and 120 min).

**Figure 8 ijms-25-00528-f008:**
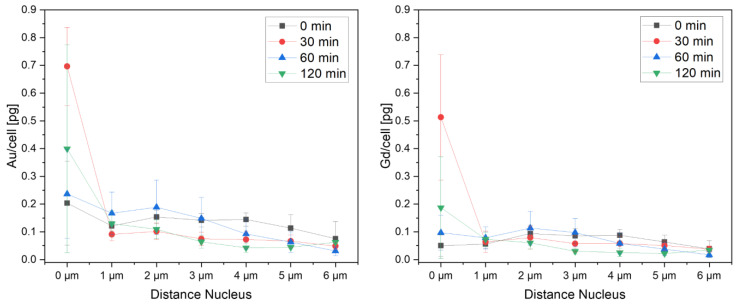
Elemental amount of Au (**left**) and Gd (**right**) according to the distance d_nuc_ to the cellular nucleus in 3T3 cells exposed to AuNP@CTPR-Gd (*C*_Au_~10 µg/mL) followed by different incubation times in NP-free medium: t_inc_ = 0 min—black, 30 min—red, 60 min—blue, and 120 min—green). Individual plots for each incubation time and element can be found in Appendix A.

## Data Availability

The XRF-imaging data presented in this study are openly available in the European Synchrotron Radiation Facility at (https://data.esrf.fr/doi/10.15151/ESRF-ES-450250995), accesed on 6 July 2021. The rest of the data are contained within the article.

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
