# Peer review of "Probing the Cellular Fate of the Protein Corona around Nanoparticles with Nanofocused X-ray Fluorescence Imaging"

_ijms, 2023, doi:10.3390/ijms25010528_

Round 1

Reviewer 1 Report

Comments and Suggestions for Authors

see attached file

Reviewer 2 Report

Comments and Suggestions for Authors Skiba et al. exploring the potential of X-ray fluorescence imaging (XFI) to study the intracellular fate of engineered protein coronas pre-formed on gold nanoparticles. The concept is novel and the technical capabilities demonstrated hold promise for gaining new biological insights using XFI. However, there are some issues that should be addressed to strengthen the work:   Major points: 1. The small sample size of only 3 cells per condition makes it very difficult to draw definitive conclusions, as acknowledged in the paper. While understandable given the technical constraints, statistical analysis on such a small, variable sample has very limited meaning. A post-hoc power analysis estimating the sample sizes needed in future works to achieve statistical significance would add important context.   2. More details should be provided on how the protein corona was formed and characterized on the nanoparticles prior to cellular experiments. How many proteins were bound per particle? What is the stability of the corona composition? This context is important for interpreting if changes seen intracellularly are due to degradation or other processes.   3. Co-localization analysis between Au and Gd calculated from the data may be more related to overall endosome cargo rather than corona integrity on single particles. This limitation should be discussed.   Minor point: 1. Figure quality is poor, with pixelated images and faint text difficult to read. Please provide higher resolution image files. Comments on the Quality of English Language

Minor editing of English language required
